# Ultrasound-Verified Peripheral Arthritis in Patients with *HLA-B*35* Positive Spondyloarthritis

**DOI:** 10.3390/life11060524

**Published:** 2021-06-04

**Authors:** Daniela Šošo, Jure Aljinović, Sanja Lovrić Kojundžić, Ivanka Marinović, Esma Čečuk Jeličić, Daniela Marasović Krstulović

**Affiliations:** 1Department of Rehabilitation Medicine and Rheumatology, University Hospital Centre Split, 21 000 Split, Croatia; jure.aljinovic@mefst.hr (J.A.); imarinovic27@net.hr (I.M.); 2Department of Health Studies, University of Split, 21 000 Split, Croatia; 3Department of Diagnostic and Interventional Radiology, University Hospital Centre Split, 21 000 Split, Croatia; lovric.sanja@gmail.com; 4Department of Transfusion Medicine, University Hospital Centre Split, 21 000, Split, Croatia; esma.cecuk@gmail.com; 5Department of Rheumatology and Clinical Immunology, University Hospital Centre Split, 21 000 Split, Croatia; daniela.marakrst@gmail.com

**Keywords:** *HLA-B*35*, peripheral arthritis, ultrasound, *HLA-B*27*, inflammatory bowel disease, psoriasis, undifferentiated axial spondyloarthritis

## Abstract

Background: We aimed to investigate possible association between the *HLA-B*35* allele and peripheral arthritis, tenosynovitis and enthesitis. Methods: Ultrasound of peripheral joints and tendons was performed in 72 *HLA-B*35* positive patients with preliminary diagnosis of undifferentiated axial form of spondyloarthitis and joint and tendon pain. Patients with other known types of axial and peripheral spondyloarthritis were excluded as well as patients with other known types of arthritis. Results: Pathological changes were found in the joints of 33 (46%) patients and on the tendons in 13 (18%) patients. The most common ultrasound findings were joint effusion and synovial proliferation with positive power Doppler signal grade 1. The most common ultrasound finding in patients with painful tendons was tenosynovitis. A higher disease activity and an increased incidence of elevated CRP (≥5 mg/L) were more often observed in the group with positive ultrasound findings. Conclusion: In this study, we showed that the *HLA-B*35* allele could be a potential risk factor for developing peripheral arthritis, but not for tenosynovits and enthesitis in patients with the undifferentiated axial form of spondyloarthritis. This result may influence the follow up of these patients, especially since it gives us an opportunity to consider the use of different types of DMARDs in the treatment of these patients.

## 1. Introduction

Spondyloarthrtis (SpA) is a group of diseases that includes ankylosing spondylitis (AS), reactive arthritis (ReA), arthritis related to inflammatory bowel disease (IBD), psoriatic arthritis (PsA), juvenile Spa and the undifferentiated form of SpA (un-SpA) [1]. The patients with un-SpA may over time evolve to an overt form of SpA [2,3,4]. Depending on whether the spine or peripheral joints and entheses are predominantly affected by inflammation, we distinguish between axial (axSpA) [5] and peripheral SpA [6]. Inflammation can lead to ankylosis of the spine, resulting in spinal rigidity. The clinical picture of patients with peripheral SpA is characterized by peripheral joints arthritis, enthesitis and dactylitis [7,8]. SpA is also often accompanied by *HLA-B*27* positivity [9], which, along with sacroiliitis, is one of two major features of axSpA [5]. Despite the unclear mechanism, *HLA-B*27* alleles are crucial in the development of SpA [10,11]. Studies have shown that in the gut and synovial tissues, β2 microglobulin (β2m), a noncovalent part of the MHC-I complex, reduces *HLA-B*27* proper folding, thereby activating the interleukin-23/interleukin-17 (IL-23/IL-17) pathway. Activation of the IL-23/IL-17 pathway leads to inflammation of spine and peripheral joints, implying that this process is associated with intestinal inflammation [12,13]. In addition to this genetic influence, many studies have shown the importance of the gut microenvironment, and particularly the gut microbiome in SpA, so it was reasonable to assume that there is a gut–joint axis in the pathogenesis of these diseases [14,15]. IL-23/IL-17 pathway dysfunction was not only detected in SpA but also in IBD, psoriasis and rheumatoid arthritis (RA) [16]. IL-23 and IL-17 are thought to be the major cytokines for axSpA and PsA [17] and the clinical picture of these patients can be greatly mitigated by secukinumab and ixekizumab, anti-IL-17 monoclonal antibodies [18]. On the other hand, these monoclonal antibodies did not manage to reduce the severity of Chron’s disease in clinical trials. Results from some studies have suggested that these drugs could even worsen the symptoms of IBD. In addition to IL-17, tumor necrosis factor-α (TNF-α) has also been shown to be an important cytokine in the pathogenesis of these diseases. TNF-α blockade has been shown to be effective not only in SpA but also in IBD, psoriasis and uveitis [19]. 

In addition to *HLA-B*27*, some studies showed an increased incidence of some other *HLA* alleles in *HLA-B*27* negative SpA patients, especially *HLA-B*35* allele [20,21,22,23]. Kamanli et al. and Said-Nahal et al. found higher frequency of *HLA-B*35* in *HLA- B*27* negative SpA patients [24,25] and genetic research on ancient human remains showed an association of *HLA-B*40*, *HLA-B*27* and *HLA-B*35* alleles in individuals with rheumatic diseases, particular in individuals with SpA [26].

This association is based on studies in which sacroiliitis was detected by conventional X-ray but the structural lesions visible on X-ray are a sign of advanced disease and therefore diagnosing with X-rays may delay diagnosis of SpA for seven years [27]. Apart from its inability to detect early sacroilliitis, there is also significant observer variation in reading radiographs of sacroiliac (SI) joints [28]. For these reasons, and especially for the development of biologic drugs that effectively treat these diseases, there was a need for early and accurate diagnosis of SpA. Since sacroiliitis, along with *HLA-B*27* positivity, is one of the two main features of axSpA, the Assessment of SpodyloArthritis International Society (ASAS) classification criteria for axSpA [5] included acute inflammation of the sacroiliac joints seen by magnetic resonance imaging (MRI) as a feature of early sacroiliitis and only definite radiographic sacroiliitis as a feature of advanced disease [29]. 

Given that we also noticed an increased frequency of the *HLA-B*35* allele in patients with symptoms of axSpA and without any other known cause of axSpA, we conducted a study in which we confirmed the connection between sacroiliitis detected by MRI and the *HLA-B*35* allele in patients with un-axSpA [30]. 

The importance of this connection lies in the fact that patients with SpA but without *HLA-B*27* positivity, recent infections, psoriasis, or IBD often go unrecognized and inadequately treated. 

On the other hand, some authors have linked *HLA-B*35* to peripheral arthritis. Dubost et al. suggested an association between the *HLA-B*35* and peripheral arthritis, which they assumed was a new entity [31], and Moroldo et al. showed the association between *HLA-B*35* and pauciarticular-onset juvenile RA [32]. Additionally, in 2000, Orchard et al. found a connection between the *HLA-B*35* allele and peripheral arthropathy as a part of the clinical picture of SpA related to IBD [33]. Ultrasound (US) was not used to evaluate joint and tendon problems.

On the other hand, in our study [30] some *HLA-B*35* patients complained of pain and swelling of the joints and tendons or these symptoms were detected by a physician during a clinical examination. We found that the *HLA-B*35* allele was associated with a five-fold greater chance of finding peripheral joint synovitis in un-axSpa patients. We assumed that these findings were part of the clinical picture of un-axSpA and not a separate clinical entity and not exclusively a part of SpA related to IBD.

In order to investigate a possible association of *HLA-B*35* positivity and these peripheral manifestations in patients with un-axSpA, we performed US examination of painful and swollen joints and tender tendons and entheses to see if these data match the clinical picture of other known forms of spondyloarthritis.

## 2. Materials and Methods

This study was performed at the Rheumatology Outpatient Clinic at University Hospital Centre in Split, Croatia, from April 2017 to January 2019. US was performed in 72 *HLA-B*35* positive patients with preliminary diagnosis of un-axSpA and joint and tendon problems. These patients were part of our cross-sectional study that showed an association between sacroiliitis detected by MRI and *HLA-B*35* allele after patients with *HLA-B*27* allele, psoriasis, IBD, preceding infections and juvenile SpA were excluded [30]. Detailed history taking and additional laboratory analysis were performed to rule out other types of arthritis.

The level of disease activity was determined by the Disease Activity Score 28 ESR (DAS28ESR) and Disease Activity Score 28 CRP (DAS28CRP) [34]. The level of functional limitation was determined by the Health Assessment Questionnaire (HAQ) [35]. DAS 28ESR, DAS28CRP and HAQ are the instruments for evaluating and monitoring patients with peripheral arthritis. They were included in daily clinical practice and in clinical research after being extensively validated.

HLA genotyping was performed at the Department of Transfusion Medicine, University Hospital Centre in Split, Croatia. Genomic DNA was extracted from EDTA-blood samples using the High Pure PCR Template Preparation Kit (Roche Diagnostics GmbH, Germany). *HLA* alleles were detected by the PCR-sequence specific oligonucleotide probing method, using the commercially available Immucor Lifecodes HLA-SSO typing kit (Immucor Transplant Diagnostics, Inc, Stamford, CT, USA) and the standard polymerase chain reaction sequence-specific priming protocol for Olerup SSP^®^ typing kits (Olerup GmbH, Vienna, Austria) [36]. HLA typing was performed using Luminex^®^ 100/200ᵀᴹ System analyzer (Luminex Corporation, Austin, TX, USA).

Ultrasonography: All patients with joint and tendon problems underwent US examination (LOGIQ e, General Electric, Shanghai, China). US examinations were performed by a single reader (J.A.) according to European League Against Rheumatism (EULAR) guidelines [37]. The reader passed Level 2 of the Competency Assessment in Musculoskeletal Ultrasound. He is also a qualified teacher of the basic and intermediate EULAR course [38]. The intra-observer variability was checked; ten randomly selected US findings were re-analyzed by the same reader.

Data analysis: Statistical analysis was performed using SPSS 20 statistical package (IBM Corp, Armonk, NY, USA). We used the Kolmogorov–Smirnov test to determine whether quantitative data follow a normal distribution. Since the data did not follow the normal distribution, we used the non-parametric Mann–Whitney U test [39]. We also used χ^2^ and logistic regression, and the results with *p* < 0.05 were considered statistically significant.

## 3. Results

Demographic and baseline characteristics of the 72 enrolled patients are shown in Table 1. 

In our group of 72 *HLA-B*35* positive patients, 57 (79%) complained of pain and swelling of the joints and 14 (19%) of pain and swelling of tendons, or these symptoms were detected by a physician during a clinical examination. Positive US findings were found in the joints of 33 (46%) patients and on the tendons in 13 (18%) patients. Most patients had the asymmetric and oligoarticular form of the disease. One third of the 72 *HLA-B*35* positive axSpA patients had grade 1 hypertrophic synovitis (32%) on the B mode scale (Figure 1) However, when those patients were scanned in PD (power Doppler) mode a substantial number of grade 2 and grade 3 PD signal was found in these patients (26% and 13%, respectively). If we take into account all 72 patients, then 8% of them had a grade 2 PD signal and 4% of them had a grade 3 PD signal. Six of the 13 patients with erosions had only inactive erosions without signs of arthritis. The wrists, metacarpophalangeal (MCP), proximal (PIP) and distal interphalangeal (DIP), and knees were most commonly affected, accounting for 12 (17%), 11 (15%), 8 (11%), 8 (11%) and 10 (14%) patients, respectively. The most common US finding in patients with painful tendons was tenosynovitis (Figure 1). 

We have shown that patients with positive ultrasound findings have a higher level of CRP and a higher level of disease activity (*p* < 0.05), but we cannot say with certainty that this is indeed the case because the 95% confidence interval (CI) includes the null value (Table 2). 

Although higher CRP levels were more often observed in the group with positive US findings, there was no statistically significant association between elevated acute phase reactant levels (ESR ≥ 29 mmHg and CRP ≥ 5 mg/L) and positive US findings (*p* > 0.05) (Table 3).

There was no statistically significant association between bone marrow oedema (BMO) at SI joints, the main criteria for diagnosis of sacroiliitis on MRI findings, and the presence of arthritis (χ^2^ = 1.51; *p* = 0.019), tenosynovitis (χ^2^ = 0.031; *p* = 0.186) or enthesitis. 

## 4. Discussion

The association between *HLA-B*35* allele and SpA, whether in the axial or the peripheral form of the disease, has long been known [20,21,22,23,24,25,26,31,32,33]. Indeed, in our study, a large proportion of the *HLA-B*35* positive patients with preliminary diagnosis of un-axSpA had joint and tendon pain and swelling according to patients or their physicians [30], which we evaluated by US. In current practice in rheumatology, US is used to evaluate peripheral joint and periarticular abnormalities, whether to confirm or refute them [40] or to detect their subclinical presence, which would otherwise be missed on examination [41]. This is especially important when clinical signs are sparse and serological tests fail to conclusively differentiate between some types of arthritis. The data on the number of joints involved, the size of the joints involved (large or small) and the pattern of joint involvement (symmetrical or asymmetrical), although sometimes similar in different types of arthritis [42], can still help decide which type of arthritis a particular patient has. In our patients with undifferentiated form of axSpA, US detected mostly asymmetric and oligoarticular arthritis of the lower limbs, which would correspond to the clinical picture of other types of SpA, including AS, ReA, and type 1 arthritis related to IBD for which association with the *HLA-B*35* allele has been described [33,43]. The high proportion of hand involvement could be explained by the fact that some forms of the SpA, such as type 2 arthritis related to IBD and the psoriatic form of SpA, can affect the hands [44,45]. Most of our patients had grade 1 hypertrophic synovitis with PD showing mild alteration grading 1. It can be argued that some patients with the synovial hypertrophy grade 1 and PD grade 1 can be found in a healthy population [46] but our study design tried to limit this accidental finding by examining only the patients with pain and swelling. This fact gives rise to the possibility that some subclinical synovitis were not detected. We consider this as a possible limitation of this study since some studies showed that SpA patients can have subclinical synovitis [47]. On the other hand, we had patients with PD signal grade 2 and 3, which is uncommon or absent in healthy subjects or in patients with osteoarthritis.

Although the correlation between mild peripheral arthritis and the *HLA-B*35* allele has already been described by Dubost et al., they had not confirmed it by US [31], nor had Orchard et al. in patients with SpA related to IBD [33]. Additionally, Dubost et al. suggested that this arthritis is a separate clinical entity but we assume that it is more likely part of the clinical picture of SpA, whether as a part of peripheral SpA or as a part of axSpA. Anti-citrullinated protein antibody (ACPA) and rheumatoid factor (RF) were detected in a few patients, but none of them fulfilled the classification criteria for RA nor for other known types of arthritis [48]. Only slightly higher disease activity and slightly higher levels of CRP were found in patients with a positive US finding. The reason lies in the fact that patients suffering from a peripheral form of SpA, in contrast to patients suffering from RA, for example, are less likely to have destructive forms of arthritis accompanied by high inflammatory parameters [49]. 

Only a few patients complained of tendon problems and US mostly showed tenosynovitis. Enthesitis was detected in two patients only. Tenosynovitis and enthesitis were expected given that they are the key respective sonopathologies in SpA [50] but the small number with these findings was unexpected. In our opinion, the reason for this lies in the fact that we evaluated only tender sites. Additionally, some of our patients were obese and enthesitis in obese patients is often difficult to diagnose clinically [51]. It is possible that due to obesity we detected a smaller number of enthesitis cases and consequently, fewer enthetes were evaluated by US. In addition, it is known that these patients have a significant prevalence of asymptomatic musculoskeletal involvement [52] and although there are some disagreements as to which entheses should be examined, these sites must be evaluated whether they are painful or not. The fact that this was not done we consider also to be a limitation of this study. Additionally, all US examinations were performed by a single examiner blinded from the clinical data. He was a unique expert in the field in our institution according to the competency of Assessment in EULAR Musculoskeletal Ultrasound. For this reason, the inter-observer variability was not determined. Therefore, the fact that only one examiner performed US and that no intra-observer variability was determined is considered a limitation of the study. In our study the patients were mostly female, which is not common for spondyloarthritis, so this could be a subject of further research. 

In rheumatology, the presence of IBP and pain in the joints and tendons always raises the suspicion of the presence of SpA. The presence of the *HLA-B*27* allele, psoriasis, IBD, or recent infection in these patients facilitates the diagnosis of this disease. In the absence of these features, patients often go unrecognized and inadequately treated.

Given that we have shown that the *HLA-B*35* allele could be a potential risk factor for developing sacroiliitis as well as for peripheral arthritis in patients with un-axSpA, we believe that the presence of the *HLA-B*35* allele justifies a complete rheumatological assessment just like patients with differentiated forms of the disease. Yet, since the patients with un-SpA may over time evolve to an overt disease, more specifically into PsA and arthritis related to IBD, the presence of *HLA-B*35* should, in rheumatologists, raise awareness of the possible development of psoriasis and IBD or even the existence of subclinical inflammatory lesions in the gut. This is of particular importance because it is known that up to 70% of AS patients have clinically silent gut inflammation, detected by ileocolonoscopy, and 7% of them will develop IBD [51]. It is possible that disruption of the epithelial intestinal barrier is likely to alter both the local and systemic inflammatory response representing the mechanism of SpA. On the other hand, some studies have suggested that anti-IL-17 blockers could worsen the symptoms of IBD. Therefore, the possibility that these patients may develop IBD or that they already have clinically silent gut inflammation should raise caution with rheumatologists when choosing a biologic drug given the tissue-specific roles of IL-17, which are probably detrimental in the joints and protective in the gut. Treatment with anti-TNF blockers might be a better choice for some of these patients.

In conclusion, although we have come a long way in understanding the pathogenesis of SpA, in everyday clinical practice we have, in addition to the clinical picture and imaging, only help from HLA typing in diagnosing these diseases. Analysis of *non-HLA* gene involvement and the complex process of antigen processing and presentation are only available in scientific studies. Therefore, early implementation of MRI in detection of sacroiliitis and US in detection of arthritis, enthesitis and dactylitis in *HLA-B*35* positive patients with a preliminary un-axSpA diagnosis is justified to better estimate the prevalence of SpA. The presence of the *HLA-B*35* allele in un-axSpA patients can also raise awareness of the possible development of psoriasis and IBD and help in the choice of therapeutic modalities. Given the very effective modalities of treatment of these diseases, early initiation of treatment with targeted therapies can prevent the development of joint damage and disability and increase patients’ quality of life. Additionally, a multidisciplinary approach, i.e., cooperation between rheumatologists, gastroenterologists, dermatologists and radiologists, is needed for monitoring and treatment of these patients as well as for further research in understanding the mechanism of these diseases and their more effective treatment.

## Figures and Tables

**Figure 1 life-11-00524-f001:**
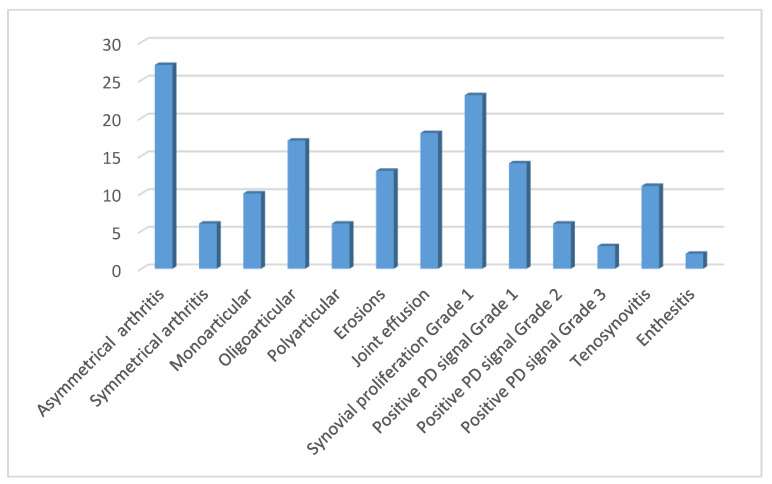
Patients (n) with positive US finding in the group of 72 *HLA-B * 35* SpA patients. Abbreviations: US: ultrasound; SpA: spondyloarthritis, PD: power Doppler.

**Table 1 life-11-00524-t001:** Demographic and baseline characteristics in 72 *HLA-B*35* positive patients.

Age, years	Median (IQR)	56 (51–65)
Female	n (%)	61 (85)
Disease duration, months	Median (IQR)	14 (9–19)
ESR (mm/h)	Median (min–max)	10 (1–60)
ESR ≥ 29 mm/h	n (%)	11(15)
CRP (mg/L)	Median(min–max)	3(0.3–42)
CRP ≥ 5 mg/L	n (%)	15 (21)
Joint pain and swelling	n(%)	57(79)
Tendon pain and swelling	n (%)	14 (19)
Dactylitis	n(%)	3 (4)
ACPA	n (%)	2 (3)
RF	n (%)	3 (4)
ANA	n (%)	
NSAIDs	n (%)	65 (90)
csDMARDs	n (%)	30 (42)
CSs	n (%)	15 (21)
bDMARDS	n (%)	3 (4)

Abbreviations: ESR: erythrocyte sedimentation rate; CRP: C-reactive protein; ACPA: anti-citrullinated protein antibody; RF: rheumatoid factor; ANA: anti-nuclear antibody; NSAID: non-steroidal anti-inflammatory drug; csDMARD: conventional synthetic disease-modifying anti-rheumatic drugs; CS: corticosteroids; bDMARD: biological disease-modifying anti-rheumatic drugs.

**Table 2 life-11-00524-t002:** Measures of disease activity and physical functioning in patients with and without a positive US finding.

72 *HLA-B*35* Positive Patients					
		Positive US (n = 33)	Negative US (n = 39)	*p*-Value ^†^	Difference (95% CI)
ESR (mm/h)	Median (IQR) (min–max)	9 (5–15) (1–60)	10 (5–18) (1–52)	0.743 ^†^	
CRP (mg/L)	Median (IQR) (min–max)	4 (1.8–7) (0.8–42)	2.9 (1–3.7) (0.3–15)	0.039 ^†^	1.1 (−0.38–2.6)
DAS28ESR	Median (IQR) (min–max)	4.3 (3.1–4.8) (1.8–5.8)	3.5 (2.5–4.1) (0.4–5.7)	0.039 ^†^	0.8 (−0.1–1.72)
DAS28CRP	Median (IQR) (min–max)	4.1 (3.4–4.6) (2.3–5.8)	3.4 (2.4–3.9) (1.2–4.9)	0.002 ^†^	0.7 (−0.03–1.31)
HAQ	Median (IQR) (min–max)	1.25 (1–1.7) (0–2.5)	1.1 (0.9–1.4) (0.4–2.3)	0.124 ^†^	

^†^ Mann–Whitney U test.

**Table 3 life-11-00524-t003:** The association between acute phase reactants and positive US findings.

72 *HLA-B*35* Positive Patients					
		Positive US (n = 33)	Negative US (n = 39)	*p*-Value *	OR(95%CI) *p*-Value **
ESR (≥29 mmHg)	n (%)	4 (12)	5 (13)	0.955	0.76 (0.19–3) 0.691
CRP (≥5 mg/L)	n (%)	9(27)	6(15)	0.344	2.1 (0.65–6.6) 0.221

* χ^2^ test; ** logistic regression.

## Data Availability

The data presented in this study are available on request from the corresponding author.

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
