# Peer review of "Ultrasound-Verified Peripheral Arthritis in Patients with HLA-B*35 Positive Spondyloarthritis"

_life, 2021, doi:10.3390/life11060524_

Round 1

Reviewer 1 Report

Comments to the Authors: Manuscript ID: life-1241525
Title: “Ultrasound-verified peripheral arthritis in patients with HLA-B*35 positive spondyloarthritis”.

The authors presented a research assessing a possible association between the HLA-B*35 allele and peripheral arthritis, tenosynovitis and enthesitis in patients with undifferentiated axial spondyloarthritis (un-axSpA).

In the previous study, in the same group of patients, the Authors confirmed the connection between sacroiliitis detected by MRI and HLA80 B*35 allele in patients with un-axSpA. 

The topic is interesting, however I have some major comments for the Authors:

  1. The group presented in the study includes many females (85%), which is not typical for axSpA. It might be due to HLA-B*35 positivity, however it has not been discussed
  2. The group presented in the study consists of patients with inactive disease (increased inflammatory parameters in 15-20%).
  3. What does it mean (line 150-151) “joint problems” and “tendon problems”, “pathological changes”? It is not a professional expression. An accurate description of symptoms is required.
  4. In Table 1 it is presented that 79% of patients reported “arthralgia”. What about symptoms of joint inflammation (swelling, tenderness on palpation), it has not been presented. Arthralgia is not arthritis.
  5. In “Methods” (line 110-111) it is written, that “The level of disease activity was determined by the Disease Activity Score 28 ESR 111 (DAS28ESR) and Disease Activity Score 28 CRP (DAS28CRP)”. However, DAS28 is calculated with the number of 28 specific joints, which do not include DIP joints. DIP joints have been reported to be affected in patients included in the study. It is possible, that joints of lower limbs have been also affected and not included in DAS28. This method of disease activity assessment is not correct in patients with SpA.
  6. In Table 1 the result of ANA is missing.
  7. The title of Table 2: US features of peripheral manifestation in 72 HLA-B*35 positive SpA patients is incorrect, because data presented in Table 2 include both clinical and ultrasonography (US) data
  8. Data on US findings are basic, for example no grading of synovial proliferation (only in 32% of patients). Positive power Doppler signal was found in 31% of patients and mostly it was Grade 1 (19%) which is mild and may occur even in healthy persons or patients with osteoarthritis.
  9. There is no information, if US changes were associated with clinical symptoms
  10. The Authors reported higher level of CRP in patients with US findings, however the significance was low. No significance with ESR.     
  11. Peripheral joints involvement (asymmetrical, oligoarticular) is typical for SpA, it is no new information. In my opinion, an association with HLA-B*35 positivity is questionable according to the presented results.   
  12. I do suggest comparison of the two groups of patients: HLA-B*35 positive and HLA-B27 positive. It might be interesting.

Author Response

Author No. 1.

I sincerely thank you for all the comments. We accepted most of them. We also explained in detail the reasons why we did not take into account some remarks.

  1. The group presented in the study includes many females (85%), which is not typical for axSpA. It might be due to HLA-B*35 positivity, however it has not been discussed.

This fact is true for patients with ankylosing spondylitis (male: female ratio 3:1) but not so much for other types of spondyloarthritis; psoriatic arthritis (1:1), reactive arthritis (1) and enteropathic arthritis (1:1). The fact that women predominated in our group could be a topic for further research.

Nevertheless, we have emphasized this fact in the Discussion: line 243

  1. The group presented in the study consists of patients with inactive disease (increased inflammatory parameters in 15-20%).

ESR and CRP are elevated in a maximum of 50% of cases and are not a reflection of disease activity because these diseases often progress in parallel with normal inflammatory parameters. Also, even when elevated, they are never as high as in other inflammatory arthritis such as rheumatoid arthritis. EULAR task force mention ‘elevated CRP’ only as being the strongest predictor of a good response to TNFi therapy.

  1. What does it mean (line 150-151) “joint problems” and “tendon problems”, “pathological changes”? It is not a professional expression. An accurate description of symptoms is required.

You are absolutely right so we have corrected that: line 91-94, 99, 100, 151-152, 185 and in Table 1.

  1. In Table 1 it is presented that 79% of patients reported “arthralgia”. What about symptoms of joint inflammation (swelling, tenderness on palpation), it has not been presented. Arthralgia is not arthritis.

You are absolutely right that we did not specify exactly what we meant so we have corrected that in Table 1.

  1. In “Methods” (line 110-111) it is written, that “The level of disease activity was determined by the Disease Activity Score 28 ESR 111 (DAS28ESR) and Disease Activity Score 28 CRP (DAS28CRP)”. However, DAS28 is calculated with the number of 28 specific joints, which do not include DIP joints. DIP joints have been reported to be affected in patients included in the study. It is possible, that joints of lower limbs have been also affected and not included in DAS28. This method of disease activity assessment is not correct in patients with SpA.

DAS 28 ESR and DAS28CRP are indices of disease activity used in patients who have affected peripheral joints as part of spondyloarthritis (not only in rheumatoid arthritis) (BASDAI, ASDAS…are used to assess disease activity on the axial skeleton). Of course, they are not ideal, they do not cover all joints, but they are extensively validated and confirmed. They also do not consume much time and they are used in clinical practice and in large clinical studies of the effectiveness of different types of drugs on peripheral joints. There are other validated tests that again are not ideal, but for some we had to opt for and we chose those tests.

  1. In Table 1 the result of ANA is missing.

You are right, but as we did not have patients with a positive ANA test, we did not give a number. Now we put zero in the table.

  1. The title of Table 2: US features of peripheral manifestation in 72 HLA-B*35 positive SpA patients is incorrect, because data presented in Table 2 include both clinical and ultrasonography (US) date.

We understand why the confusion occurred and we would to explain further. In Table 1, we listed the number of patients who complained of painful and swollen joints or painful or swollen tendons during the clinical examination, or who had these problems according to the doctor's clinical examination. Therefore, only the data obtained by ultrasound are listed in Figure 1 (the other reviewer asked to make Table into a bar graph). So  we've clarified "joint and tendon problems" as you requested, in the tables and in the text (line 91-94, 99, 100, 151-152, 185 and in Table 1)

  1. Data on US findings are basic, for example no grading of synovial proliferation (only in 32% of patients). Positive power Doppler signal was found in 31% of patients and mostly it was Grade 1 (19%) which is mild and may occur even in healthy persons or patients with osteoarthritis.

All of our patients had grade 1 hypertrophic synovitis (in B mode), which we have now further emphasized

It is true that in the healthy subjects hypertrophic synovitis can be found but PD signal showed mild alteration grading 1 (out of 3) but  lesions scoring 2 or 3 were uncommon or absent. And in our study, we had patients with PD signal 2 (8%) or 3 (4%) which is a sign of neovascularization and inflammatory arthritis. We must emphasize that this is the percentage we counted on all HLAB*35 SpA patients and not on those we examined with ultrasound so we think this is not a negligible figure especially since some studies showed that SpA patients frequently have subclinical synovitis.

But, you are quite right that it is necessary to state that, so we have stated that in  Figure 1 and stated in Results section line 154-163 and in Discussion (line 202-210) and  elaborated on this in more detail.

  1. There is no information, if US changes were associated with clinical symptoms.

Further explanation: We stated, and pointed out this as a limitation of the study, that we only ultrasounded the joints and tendons that had a clinical presentation according to patient or his physician.

  1. The Authors reported higher level of CRP in patients with US findings, however the significance was low. No significance with ESR.

This is truth and we stated out this in the article: We have shown that patients with positive ultrasound findings have higher level of CRP and higher level of disease activity (p<0.05) but we cannot say with certainty that this is indeed the case because 95% confidence interval (CI) includes null value. (Table 3).

          Our research showed such data and as such we presented them. An explanation for this (as we have

            explained in the article) can be found in the answer to your comment number 2:

  Also, we stated that the significance was low and what is needed to confirm it:

Also, to confirm the obtained statistically significant association of elevated CRP values and elevated disease activity with a positive US finding, we should have included a larger number of patients”

  1. Peripheral joints involvement (asymmetrical, oligoarticular) is typical for SpA, it is no new   information. In my opinion, an association with HLA-B*35 positivity is questionable according to the presented results.   

We agree with you that this pattern has been recognized before and we have highlighted this fact in the article. As we also stated in the article, we wanted to objectively, ie by ultrasound (which has not been done so far) prove whether there is arthritis and enthesitis in a patient with undifferentiated form of spondyloarthritis, in this case HLA-B * 35 + and whether there is a similarity and connection with other, differentiated forms of spondyloarthritis. Namely, in current rheumatological practice it is necessary to evaluate the peripheral joint with ultrasound because there is synovitis (exudative or hypertrophic that are not clinically visible) in patients who present only with pain as well it is necessary to confirm with ultrasound clinically diagnosed joint swelling (it is known that clinical assessment may be wrong and sometimes impossible as for example at the shoulder joint).

  1. I do suggest comparison of the two groups of patients: HLA-B*35 positive and HLA-B27 positive. It might be interesting.

I agree. A comparison of the ultrasound findings of these two groups of patients could be interesting and I think this is an excellent topic but for some further research.

Best regards and once again a sincere thank you for all the comments!

Daniela Šošo

Reviewer 2 Report

Dear Authors

Please take a look at the following:

  • Please explain why you used IQR or max-min or provide all measures for all variables
  • Please make table 2 into a bar graph
  • In table 3 use . and not , (suggested to make a scatter plot for this table and table 4)
  • Please combine small tables to make a big one
  • Please perform a logistic regression to report the odds ratio

Best Regards

Author Response

I sincerely thank you for all the comments

Please take a look at the following:

  • Please explain why you used IQR or max-min or provide all measures for all variables

               With the help of our statistician, we changed the IQR to Q1-Q3 and provided measurements for all variables. If you deem it necessary, we can remove one of the two variables (Q1-Q3 or min-max)

  • Please make table 2 into a bar graph.

               We did it as you requested (Figure 1.)

  • In table 3 use . and not , (suggested to make a scatter plot for this table and table 4)

We changed as you requested.

We apologize for not making a scatter plot but we think that in this way valuable data will be lost.

  • Please combine small tables to make a big one

  • Please perform a logistic regression to report the odds ratio

               Answer to the last two remarks:

We merged the last two tables into one, but when we performed the logistical regression you were looking for, the table turned out to be too wide.

So we separated them though.

Since there was no statistically significant difference, we did not make a logistic regression earlier, but now we made it according to your request.

Thanks again to all your remarks and best regards!

Daniela Šošo

Reviewer 3 Report

I think the work is original and interesting for publication. It would be necessary to increase the introduction and provide some explanatory image.

Author Response

I sincerely thank you for all the comments.

It would be necessary to increase the introduction and provide some explanatory image.

In the first version of the article, we had a significantly longer introduction, but we received an explicit request from the previous reviewer to shorten the introduction, which was accepted. Therefore, please accept this introduction or have the editor return a longer version in agreement with you.

This was our first and for about 300 words longer version:

Spondyloarthrtis (SpA) is a group of diseases which includes ankylosing spondylitis (AS), reactive arthritis (ReA), arthritis related to inflammatory bowel disease (IBD), psoriatic arthritis (PsA), juvenile Spa and undifferentiated form of SpA (un-SpA) [1]. The patients with un-SpA do not meet diagnostic criteria of other types of SpA, but may over time evolve to an overt form of SpA [2-4]. In addition to IBD and psoriasis, extra-articular manifestations of SpA include also uveitis which shows their common genetic predisposition. Depending on whether the spine or peripheral joints and entheses are predominantly affected by inflammation, we distinguish between axial (axSpA) [5] and peripheral SpA [6]. Inflammation of the spine, respectively sacroiliitis and spondylitis, is typically manifested by inflammatory back pain (IBP). In advanced cases this inflammation leads to ankylosis of the spine resulting in spinal rigidity. The clinical picture of patients with peripheral SpA is characterized by peripheral joints arthritis, enthesitis and dactylitis with enthesitis [7] as the key pathological feature of SpA [8]. SpA is also often accompanied with HLA-B*27 positivity [9] which, along with sacroiliitis, is one of two major features of axSpA [5]. HLA-B*27 is major histocompatibility complex (MHC) class I allele. The genetic profile of HLA-B*27 includes HLA-B*2705, *2707, *2708, and *2710 alleles [10]. Despite the unclear mechanism, HLA-B*27 allels are crucial in the development of SpA, especially of AS and almost 95% of patients with AS are HLA-B*27 positive [11]. The studies have showed that β2 microglobulin (β2m), a noncovalent part of the MHC-I complex, reduces HLA-B*27 proper folding thereby activating interleukin-23/interleukin-17 (IL-23/IL-17) pathway. Activation of the IL-23/IL-17 pathway leads to inflammation of spine and peripheral joints implying that this process is associated with intestinal inflammation [12,13].  Also, in addition to this genetic influence, many studies have shown the importance of the gut microenvironment, particularly, the gut microbiome in SpA so it was reasonable to assume that there is a gut-joint axis in pathogenesis of these disease [14,15]. IL-23/IL-17 pathway dysfunction was not only detected in SpA but also in IBD, psoriasis and rheumatoid arthritis (RA) [16]. IL-23 and IL-17 are thought to be the major cytokines for axSpA and PsA [17] and it has been shown that the clinical picture of these patients can be greatly mitigated by secukinumab and ixekizumab, anti-IL-17 monoclonal antibodies [18]. On the other hand, these monoclonal antibodies did not manage to reduce severity of Chron´s disease in clinical trials. Results from some studies have suggested that these drugs could even worsen the symptoms of IBD but have not shown a significant increase in de novo IBD development. In addition to IL-17, tumor necrosis factor-α (TNF-α) has also been shown to be an important cytokine in pathogenesis of these disease. TNF-α blocade has been show effective not only in SpA but also in IBD, psoriasis and uveitis [19].

Apart from MHC, two non-MHC genes, interleukin 23 receptor (IL-23R) and endoplasmic reticulum amino-peptidase 1 (ERAP1) have been identified in patients with AS [20,21]. 

So far, no studies have shown a strong association of non-B*27 genes with SpA but some studies have shown possible association of HLA-B*08, HLA-B*38, HLA-B*39 and HLA-C*0602 with PsA [22]. In addition to HLA-B*27, some studies showed increased incidence of HLA-B*7, HLA-B*16, HLA-B*35, HLA-B*38 and HLA-B39 allele in HLA-B*27 negative SpA patients [23-26]. In 2009, an increased incidence of HLA-B*35 allele in HLA-B*27 negative SpA patients was noted by Kamanli et al. [27]. Said Nahal et al. found that the frequency of HLA-B*35 was higher in HLA- B*27 negative SpA patients in 117 French families [28]. Genetic research on ancient human remains in a medieval necropolis have shown an association of HLA-B*40, HLA-B*27 and HLA-B*35 alleles in individuals with rheumatic diseases, particular in individuals with SpA [29].

This association is based on studies in which sacroiliitis was detected by conventional X-ray. Unfortunately, the structural lesions visible on X-ray typical for radiographic sacroiliitis are a sign of advanced disease and therefore diagnosing with X-rays may delay diagnosis of SpA for 7 years [30]. Apart from its inability to detect early sacroilliitis, there is also significant observer variation in reading radiographs of sacroiliac (SI) joints [31]. For these reasons, and especially for the development of biologic drugs that effectively treat these diseases, there was a need for early and accurate diagnosis of SpA. Since sacroiliitis, along with HLA-B*27 positivity, is one of the two main features of axSpA, the Assessment of SpodyloArthritis International Society (ASAS) classification criteria for axSpA [5] included acute inflammation of the sacroiliac joints seen by magnetic resonance imaging (MRI) as a feature of early sacroiliitis and only definite radiographic sacroiliitis as a feature of advanced disease [32]. 

Given that we noticed an increased frequency of HLA-B*35 allele in patients with symptoms of axSpA and without any other known cause of axSpA and given all the above, we wanted to investigate whether there is a connection beetwen HLA-B*35 allele and un-axSpA in our patients. To confirm the thesis that HLA-B*35 allele is associated with the possible development of axSpA, we conducted a study in which we confirmed the connection between sacroiliitis detected by MRI and HLA-B*35 allele in patients with un-axSpA [33].

               The importance of this connection lies in the fact that patients with SpA but without HLA-B*27 positivity, recent infections, psoriasis, or IBD often go unrecognized and inadequately treated. Therefore, the presence of HLA-B*35 allele, even when there are no other known causes of SpA, should raise awareness of the existence of SpA and of the need for a full diagnostic work-up and further monitoring.

However, in 1999. Dubost et al. suggested an association between the HLA-B*35 and peripheral arthritis they assumed was a new entity [34]. In 2004., Moroldo et al. confirmed an earlier thesis of the association between HLA-B*35 and pauciarticular-onset juvenile RA [35]. Also, in 2000., Orchard et al. found connection between the HLA-B*35 allele and peripheral arthropathy as a part of the clinical picture of SpA related to IBD [36]. These studies have shown an association of HLA-B*35 and mild peripheral arthritis, but no ultrasound (US) was used to evaluate joint and tendon problem.

On the other hand, a large proportion of the HLA-B*35 positive un-axSpA patients in our study had joint and tendon problems, which we assumed were part of the clinical picture of un-axSpA and not a separate clinical entity and not exclusively a part of SpA related to IBD.

In order to investigate a possible association of HLA-B*35 positivity and these peripheral manifestations in patients with un-axSpA, we performed US examination of painful and swollen joints and tender tendons and entheses.

Also, we added an additional Figure. Ultrasound findings are available on request.

Best regards!

Daniela Šošo

Round 2

Reviewer 1 Report

The corrected version of the manuscrilt is a acceltable fot publication.

This manuscript is a resubmission of an earlier submission. The following is a list of the peer review reports and author responses from that submission.

Round 1

Reviewer 1 Report

Your study aimed to investigate possible association between the HLA-B*35 allele and peripheral arthritis, tenosynovitis and enthesitis.

Major revision:

: please write your limitation clearly through your research.

: please shorten your introduction that is too long.

: clarify your strength points clearly.

Minor revision:

: at Table 3, you set ESR and CRP as binary variable, on the other hand, the others set as continuous variables.

Please clarify the reason.

: to association, P value is adequate? The odds ratio is used as a measure of the association between two variables. Please select an acceptable statistical method to clarify your purpose.

Author Response

Please write your limitation clearly through your research.

Dear Sir, I have clarified the restrictions (within the discussion the restrictions are written in red).

Please shorten your introduction that is too long.

I completely agree that the introduction is too long. In the original version, it was much shorter, but as it was necessary for the article to have 3000 words and for the journal to be intended primarily for gastroenterologists (not rheumatologists), the introduction ended up being too long.

Still I shortened it by about 300 words, I hope that’s okay.

Clarify your strength points clearly.

By shortening the introduction and adding text in the conclusions (written in red letters), I hope the strength is better explained.

Minor revision:

As for the statistics, you are absolutely right here (our statistician, who unfortunately did not see the finished tables, also completely agrees with you). Therefore, Tables 2 and 3 have been corrected (in red letters) and Table 4 has been added. More text has been added to the results section (in red letters).

In short, really thank you for the comments. You have read the article in detail and your comments have contributed a lot to its improvement.

Best regards,

Daniela Šošo

Reviewer 2 Report

The authors presented a valuable work about HLA-B*35 positive patients with preliminary diagnosis of un-axSpA and problems with joints and tendons, where they showed how early implementation of ultrasound can be justified to better estimate the prevalence of SpA.

However, the title is describing only "peripheral manifestation" which can be mistaken with the manifestation other than just those in joints and tendons.

Please rephrase the title and add joints and tendons if only these are involved in your examinations.

Author Response

Dear sir,

thank you for your kind words.

As for the title, you are absolutely right. So I changed the title but I only put joints (arthritis) in it since ultrasound confirmed arthritis is the main strength of our study. I hope that's OK.

Unfortunately, as in rheumatology, by the term "peripheral manifestations" we usually mean arthritis, enthesitis and dactylitis, I did not think that the title could be confusing so thank you for your remark.  

Best regards

Daniela Šošo

Round 2

Reviewer 1 Report

Thank you for revision. You revised some points according to my suggestions.

But, Overall, Not scientific logic.

Major revision:

: Line 16: your purpose: “To investigate possible association between the HLA-B*35 allele and peripheral arthritis, tenosynovitis and enthesitis.”

: Line 29 : your conclusion “HLA-B*35 allele could be a potential risk factor for developing peripheral arthritis”

However, your participants is only HLA-B*35 positive. You had better to compare between HLA-B*35 (positive or negative) and peripheral arthritis (develop or non develop).